# Glomerular Endothelial Cell-Derived miR-200c Impairs Glomerular Homeostasis by Targeting Podocyte VEGF-A

**DOI:** 10.3390/ijms232315070

**Published:** 2022-12-01

**Authors:** Raluca Ursu, Nina Sopel, Alexandra Ohs, Ramesh Tati, Lisa Buvall, Jenny Nyström, Mario Schiffer, Janina Müller-Deile

**Affiliations:** 1Department of Nephrology and Hypertension, University Hospital Erlangen, Friedrich-Alexander University (FAU) Erlangen-Nürnberg, 91054 Erlangen, Germany; 2Research Center on Rare Kidney Diseases (RECORD), University Hospital Erlangen, 91054 Erlangen, Germany; 3Bioscience Renal, Research and Early Development, Cardiovascular, Renal and Metabolism (CVRM), BioPharmaceuticals R&D, AstraZeneca, 43150 Gothenburg, Sweden; 4Institute of Neuroscience and Physiology, Sahlgrenska Academy, Gothenburg University, 41390 Gothenburg, Sweden

**Keywords:** miR-200c, podocytopathies, proteinuria, VEGF-A, glomerulus, endotheliosis, zebrafish

## Abstract

Deciphering the pathophysiological mechanisms of primary podocytopathies that can lead to end-stage renal disease and increased mortality is an unmet need. Studying how microRNAs (miRs) interfere with various signaling pathways enables identification of pathomechanisms, novel biomarkers and potential therapeutic options. We investigated the expression of miR-200c in urine from patients with different renal diseases as a potential candidate involved in podocytopathies. The role of miR-200c for the glomerulus and its potential targets were studied in cultured human podocytes, human glomerular endothelial cells and in the zebrafish model. miR-200c was upregulated in urine from patients with minimal change disease, membranous glomerulonephritis and focal segmental glomerulosclerosis and also in transforming growth factor beta (TGF-β) stressed glomerular endothelial cells, but not in podocytes. In zebrafish, miR-200c overexpression caused proteinuria, edema, podocyte foot process effacement and glomerular endotheliosis. Although zinc finger E-Box binding homeobox 1/2 (ZEB1/2), important in epithelial to mesenchymal transition (EMT), are prominent targets of miR-200c, their downregulation did not explain our zebrafish phenotype. We detected decreased vegfaa/bb in zebrafish overexpressing miR-200c and could further prove that miR-200c decreased VEGF-A expression and secretion in cultured human podocytes. We hypothesize that miR-200c is released from glomerular endothelial cells during cell stress and acts in a paracrine, autocrine, as well as context-dependent manner in the glomerulus. MiR-200c can cause glomerular damage most likely due to the reduction of podocyte VEGF-A. In contrast, miR-200c might also influence ZEB expression and therefore EMT, which might be important in other conditions. Therefore, we propose that miR-200c-mediated effects in the glomerulus are context-sensitive.

## 1. Introduction

Podocytopathies are a group of glomerular diseases in which proteinuria is attributed to damage or dysfunction of podocytes. Factors that cause podocytopathies can either be intrinsic or extrinsic. Intrinsic factors are found in genetic focal segmental glomerulosclerosis (FSGS) with distinct podocyte mutations. However, more often podocytopathies are caused by extrinsic factors, such as circulating permeability factors in primary FSGS [1], podocyte autoantibodies in membranous glomerulonephritis (MGN) [2], immunological factors in minimal change disease (MCD) [3], lupus nephritis [4], IgA nephropathies (IgAN) [5] or hemodynamic factors, which are typical for obesity or diabetes with glomerular hyperfiltration [6]. In all these extrinsic causes, the glomerular endothelium primarily faces the factors that are believed to cause the podocyte damage.

We hypothesize a common pathomechanism that derives from glomerular endothelial cells and contributes to podocyte damage in proteinuric diseases.

Recently, microRNAs (miRs) were found to be involved in various diseases including kidney injury [7]. MiRs induce gene silencing by binding to the 3’UTR of its targeted mRNAs, thereby leading to mRNA degradation or inhibition of mRNA translation [8]. MiRs are stable and detectable in human biofluids, such as urine and serum, making them feasible potential biomarkers. Different urinary and blood miRs were described to be associated with distinct glomerular diseases [9]. Using our zebrafish model, we could already show the importance of miRs for glomerular integrity and identified miRs which play an important role in glomerular function [10]. Recently, we described a potential transport mechanism of glomerular endothelial cell-derived miRs to podocytes through exosomes [11].

From a comprehensive miR screening of pooled urine from patients with different glomerular diseases, as well as from different kidney cell lines in culture, we were able to identify disease as well as cell type specific miRs and could further assign the urinary miRs to different renal cell types [12]. However, each identified miR needs to be validated in further experiments.

Here we focus on miR-200c, which was upregulated in urine of different podocytopathies, namely MCD, MGN and FSGS, and could be assigned to be derived from glomerular endothelial cells after cell stress.

Overexpression of miR-200c in zebrafish by injection of an miR-200c mimic in egg stage caused edema, proteinuria, podocyte foot process effacement and glomerular endothelial cell damage. Studying potential targets of miR-200c in zebrafish and human cell culture revealed decreased VEGF-A as the most likely phenotype-causing candidate. With our study, we aim to gain more insights in podocyte injury mediated by glomerular endothelial cell-derived miRs. Furthermore, urinary miRs might become a noninvasive marker for podocyte injury in the future.

## 2. Results

### 2.1. miR-200c Is Upregulated in Urine from Patients with MCD, MGN and FSGS

We reanalyzed data from a previous miR screening in urine from patients with different renal diseases and identified miR-200c being upregulated in patients with different podocytopathies (Appendix A).

We confirmed the upregulation of urinary miR-200c in MCD, MGN and FSGS in an independent cohort of patients with different renal diseases with the help of quantitative polymerase chain reaction (qPCR) (Figure 1a). All these diseases cause severe proteinuria in patients; however, a common mechanism is unknown. MiRs are highly conserved between species. Human and zebrafish miR-200c only differ in three base pairs, while the seed sequence for mRNA binding is identical (Figure 1b). Therefore, we used the zebrafish model, which has a pronephros very similar to the human glomerulus on the structural and functional level, to investigate the glomerular effects of miR-200c.

### 2.2. miR-200c Overexpression Causes Proteinuria, Edema, Podocyte Effacement and Glomerular Endotheliosis in Zebrafish

MiR-200c mimics, as well as a control miR composed of scrambled oligonucleotides (miR-Ctrl), were injected in zebrafish larvae at the one-to-four-cell stage in various doses. Injection of miR-200c mimic induced a phenotype with the edema of the pericardium and yolk sac (Figure 2a,b). Hematoxylin and eosin (H&E) staining of the glomeruli of miR-200c mimic or miR-Ctrl-injected larvae shows fully developed glomeruli, but whole body edema was also seen in zebrafish sections with no difference in overall H&E staining (Figure 2b). We quantified the severity of the edema from P1 to P4, with P1 for fish with no edema and P4 for fish with severe edema (Figure 2c), and could show that the severity of edema correlated with the dose of the miR-200c mimic injected.

To further rule out glomerular developmental delay as a cause for edema formation, we confirmed timely fusion of the glomeruli at 48 h postfertilization (hpf) after miR-200c overexpression using Tg(wt1b:EGFP) zebrafish, which expresses eGFP under the control of the wt1b promotor (Figure 2d).

In order to differentiate between a renal or cardiac origin of the edema, we used the transgenic zebrafish line Tg(l-fabp:VDBP-EGFP) (new name Tg(fabp10a:gc-EGFP)), which constitutively expresses a circulating green fluorescent vitamin D-binding protein, roughly resembling the size of albumin. We monitored the fluorescent vitamin D-binding protein in the retinal vessel plexus [13] and observed a significant reduction in fluorescence in the miR-200c-injected fish in comparison with the miR-ctrl-injected fish. This hints to the leakiness of the glomerular filtration barrier (GFB) and loss of high-molecular-weight proteins (Figure 2e).

To assess whether the observed edema and protein loss after injection of miR-200c indeed developed because of glomerular structural changes, we analyzed the pronephros of the fish by transmission electron microscopy. We observed that zebrafish injected with the miR-200c mimic show foot process effacement and loss of endothelial fenestrations (Figure 2f). This data indicates that miR-200c could have important implications in the pathophysiology of podocytopathies.

### 2.3. miR-200c Is Induced due to Cell Stress in Glomerular Endothelial Cells

As we observed that miR-200c overexpression caused phenotypic changes in podocytes and glomerular endothelial cells in the zebrafish model, we further investigated in which of the two cell types miR-200c expression is induced. The miR screening provided evidence that miR-200c is derived from glomerular endothelial cells upon transforming growth factor beta (TGF-β) stimulation. Here, we performed independent experiments using cultured human podocytes and human glomerular endothelial cells to investigate the induction of miR-200c after stimulation with TGF-β.

We could detect a cell type-specific and time-dependent regulation of miR-200c. MiR-200c was significantly upregulated in glomerular endothelial cells after 24 h of TGF-β stimulation and decreased back to baseline after 48 h (Figure 3a). In contrast, we could not see any upregulation of miR-200c in response to TGF-β in podocytes (Figure 3b). Therefore, we conclude that TGF-β-induced cell stress causes time-dependent upregulation of miR-200c in glomerular endothelial cells.

### 2.4. miR-200c Overexpression Decreases VEGF-A in Podocytes and Downregulates ZEBs in Podocytes and Glomerular Endothelial Cells

ZEB1/2 were previously described as targets of miR-200c and as essential transcription factors playing a role in epithelial-to-mesangial transition (EMT) [14,15]. Overexpression of miR-200c downregulated ZEB1 and ZEB2 mRNA in both glomerular endothelial cells and podocytes (Figure 4a,b). However, an miR-200c-induced downregulation of ZEBs would not explain the glomerular injury seen in the zebrafish model, as reduced ZEB expression is expected to be rather protective in EMT.

In addition to ZEBs, the miR target scan revealed VEGF-A as another potential target of miR-200c (Appendix A) [16]. VEGF-A is known to be expressed in podocytes but not glomerular endothelial cells. Furthermore, we detected downregulation of VEGF-A mRNA after miR-200c overexpression in podocytes (Figure 4b).

Moreover, two orthologues, vegf-aa and vegf-ab, were downregulated in a dose-dependent manner in zebrafish after the miR-200c mimic injection in the egg stage (Figure 4c).

### 2.5. miR-200c Overexpression Decreases ZEB1 in Glomerular Endothelial Cells and ZEB1 and VEGF in Podocytes on Protein Level and Induces Cytoskeleton Rearrangement

Next, we investigated the potential downregulation of ZEB on protein level by performing immunofluorescence staining after miR-200c transfection in podocytes and glomerular endothelial cells. Unfortunately, only a ZEB1 antibody was available. Considering that ZEB1 is mainly localized in the nucleoplasm of the cells [17], we could observe a considerably diminished fluorescence signal for ZEB in the nuclei of glomerular endothelial cells and podocytes (Figure 5a,b). In accordance with the results on mRNA level, we could detect a decrease in VEGF-A in podocytes overexpressing miR-200c also on protein level (Figure 5c). Furthermore, phalloidin staining was performed to visualize intracellular F-actin stress fibers. Overexpression of miR-200c resulted in disrupted stress fibers, which were disordered. In contrast, cells transfected with an miR-Ctrl mimic exhibited regularly arranged stress fibers (Figure 5a–c). This data hints that miR-200c overexpression alters the structure of the podocyte cytoskeleton, suggesting another link to podocytopathies.

To examine if miR-200c not only decreases VEGF-A expression in human podocytes but also influences VEGF protein secretion, we measured VEGF expression after miR-200c transfection in the supernatant of podocytes. Indeed, we could observe that VEGF concentration in the cell culture supernatant was significantly lower in human podocytes transfected with miR-200c compared to control miR (Figure 5d).

## 3. Discussion

Although heterogenous as a group of diseases, podocytopathies share a malfunction of the GFB. The three layers forming the GFB are podocytes, the glomerular basement membrane and fenestrated endothelium. Damage to any of these three layers results in proteinuria. Nonetheless, podocytes are pivotal when it comes to glomerular permselectivity [18]. Increasing evidence of an intricate crosstalk between glomerular endothelial cells and podocytes through paracrine signaling is emerging as being important in physiological conditions and disrupted in disease. In addition, miR networks are involved as mediators in these signaling pathways [12].

We identified upregulation of miR-200c in urine from patients with various glomerular diseases. Interestingly, miR-200c was upregulated specifically and in a time-dependent manner in glomerular endothelial cells after stimulation with TGF-β, but not in podocytes, suggesting that miR-200c could be released from glomerular endothelial cells during injury and act in a paracrine and/or autocrine manner in the glomerulus. Recently, we were able to observe the transportation of glomerular endothelial cell-derived miRs to podocytes [11].

Here we show the detrimental effect of excessive miR-200c on podocytes and glomerular endothelial cells with disruption of actin stress fibers. Podocyte actin cytoskeleton rearrangement is the common final pathway leading to podocyte foot process effacement [19], inducing altered podocyte morphology, compromising their ability to provide adequate epithelial coverage to glomerular capillaries and resulting in proteinuria and podocyturia [20].

In line with these findings, zebrafish developed edema, proteinuria, podocyte foot process effacement and glomerular endotheliosis after miR-200c overexpression. The zebrafish is a good screening model for glomerular diseases, as the zebrafish kidney is very comparable to the human glomerular counterpart even on the ultrastructural level, and genome conservation is approximately 70% between zebrafish and humans [21]. The kidney homologue of zebrafish larvae is the pronephros, consisting of two glomeruli and two tubular systems. At 48 hpf (hours postfertilization) the glomeruli of the zebrafish pronephros fuse. A functional glomerulus is already present in zebrafish at 96 hpf, and full maturation of the GFB, with well-developed podocyte foot processes and endothelial cell fenestrations, occurs at day 4 postfertilization [22,23]. We could observe that pronephros development was not impaired in our transgenic wt1b:eGFP zebrafish line after the injection of miR-200c mimic.

Interestingly, we could also detect endotheliosis (loss of endothelial fenestrations) in zebrafish injected with a miR-200c mimic. We were interested in glomerular targets of miR-200c that mediate glomerular damage in the zebrafish model. We could show decreased VEGF-A in podocytes on the mRNA and protein level after miR-200c overexpression. Moreover, podocytes secreted less VEGF protein in the supernatant after transfection with miR-200c.

VEGF-A is required for normal development of glomerular capillaries and for the generation of their fenestrated phenotype. In addition, tightly regulated VEGF-A expression is necessary for maintaining a normal podocyte phenotype. Loss or gain of function induces diverse glomerular phenotypes in different glomerular diseases and in different stages of diseases [24,25,26,27]. VEGF-A is, on the one hand, a survival factor for podocytes and endothelial cells, while, on the other hand, excess VEGF-A, especially locally, causes extensive foot process effacement and proteinuria [28].

Considering the fact that VEGF-A has important autocrine effects on podocytes, in downregulating VEGF-A, miR-200c acts detrimentally to the homeostasis of podocytes under physiological conditions. Moreover, VEGF-A also has well-established paracrine functions, and VEGF-receptors are most abundant in glomerular endothelial cells [29,30]. Thus, through releasing miR-200c, the glomerular endothelial cells are deprived from trophic effects of VEGF-A via a negative feedback loop.

Given the downregulation of VEGF-A after the overexpression of miR-200c in cultured podocytes and the allegedly paracrine detrimental effects on glomerular endothelial cells, we speculate that the endotheliosis seen in miR-200c-injected zebrafish is caused by downregulation of VEGF. Indeed, we could show decreased vegf-Aa and vegf-Ab in zebrafish overexpressing miR-200c. Nonetheless, we could not see any blood vessel deficiencies or impairment in glomerular tuft formation. One possibility is that the level of VEGF-A downregulation by miR-200c is not severe enough to inhibit vascular development, but sufficient to induce more subtle changes in the morphology of glomerular endothelial cells. This is supported by studies in mice, where loss of both alleles of VEGF-A from podocytes leads to a marked reduction in endothelial cell migration and failure to form the GFB, whereas the loss of only one of the two VEGF-A alleles results in glomerular endotheliosis [30]. Altogether, there is a strong indication that the zebrafish phenotype induced by miR-200c is mediated by the miR-200c-VEGF axis.

In patients, miR-200c was upregulated in minimal change disease, membranous glomerulonephritis and focal segmental glomerulosclerosis, and its overexpression decreased VEGF-A in a human podocyte cell culture model. Decreased VEGF-A had been described in all of the above-mentioned podocytopathies [31,32,33,34,35,36]; however, until now a common pathomechanism was unknown. We propose that miR-200c might be the common link.

Numerous tumor-profiling studies reported that the miR-200 family is playing a central role in epithelial-to-mesenchymal transition (EMT) and that it is linked to the regulation of ZEB1 and 2, which harbors nine conserved miR-200 sites in its 3′UTR [37,38]. In addition to VEGF-A, we also identified ZEB as a potential target of miR-200c. Indeed, miR-200c overexpression decreased ZEB1 and ZEB2 in podocytes and glomerular endothelial cells. EMT induced by ZEBs is a process of reverse embryogenesis, which occurs in diseased kidneys as well as in many other organs under pathological conditions [15]. Emerging evidence indicates that the EMT of podocytes after injury is a mechanism leading to podocyte dysfunction and proteinuria [39]. The mesenchymal transition of podocytes after injury may play a vital role in causing podocyte dysfunction that ultimately leads to a defective glomerular filtration in various glomerular diseases. TGF-β is a potent EMT inducer, which is up-regulated in different proteinuric kidney diseases [40]. However, mechanisms regarding TGF-β-mediated EMT are still unknown and need further investigation.

In summary, we speculate that glomerular endothelial cell-derived miR-200c upregulated in various glomerular diseases has autocrine and paracrine effects on podocytes. Here we show that miR-200c overexpression might have detrimental effects in lowering glomerular VEGF-A signaling, which further promotes glomerular damage. In addition, upregulation of miR-200c and subsequent downregulation of ZEB1/2 in podocytes and glomerular endothelial cells might occur in a negative feedback loop to antagonize EMT, which needs further investigation. Interfering with the miR-200c-ZEB1/2 and miR-200c-VEGF-A pathway as a therapeutic option has to take into consideration that these pathways might be context-sensitive.

## 4. Materials and Methods

### 4.1. Cell Culture

Culture conditions of conditionally immortalized human podocytes were described previously [41]. In brief, podocytes were proliferated under permissive conditions at 33° C. When cultivated at 37 °C, the SV40 T-antigen was inactivated for cell differentiation. The culture medium for human podocytes was RPMI 1640 Medium (Gibcovia ThermoFisher Scientific, Waltham, MA, USA) with 10% fetal calf serum (FCS; PAN Biotech, Aidenbach, Germany), 1% penicillin/streptomycin (PS, Sigma-Aldrich Chemie GmbH, Taufkirchen, Germany)) and 0.1% insulin/transferrin/selenium (ITS, Gibco, via ThermoFisher Scientific, Waltham, MA, USA). Human glomerular endothelial cells (Clonetech, Kirkland, WA, USA) were cultivated at 37 °C and 5% CO_2_ in endothelial cell media (VascuLife^®^ Basal Media, LifeLine Cell Tech, EnGs-MV, Microvascular endothelial cell growth medium, Cell Systems, Troisdorf, Germany). This medium was supplemented with 10 mM L-glutamine LifeFactor, 5 ng/mL rh EGF LifeFactor, 1 µg/mL hydrocortisone hemisuccinate LifeFactor, 0.2% EnGS LifeFactor, 50 µg/mL ascorbic acid LifeFactor, 0.75 U/mL heparin sulfate LifeFactor, 5% FBS LifeFactor and 30 µg/mL gentamicin/15 ng/mL amphotericin B (all supplements come with the media).

Human podocytes were starved in FCS-reduced (1%) medium overnight. Then, cells were stimulated with 5 ng/mL of TGF-β (Peprotech, via ThermoFisher Scientific, Waltham, MA, USA). After stimulation, cells were harvested for RNA at 6 h, 24 h and 48 h. Unstimulated cells served as control at each time point.

#### 4.1.1. Transient Transfection of miRNA Mimics

After thermos switching to 37 °C, immortalized human podocytes were allowed to be differentiated for 8–10 days prior to transfection. Prior to transfection, cells were incubated in Opti-Mem serum-reduced medium (Gibco, via ThermoFisher Scientific, Waltham, MA, USA) for 4 h. Podocytes were transfected with mirVana^®^ miRNA mimic (hsa-miR-200c-3p from ThermoFisher (Scientific, Waltham, MA, USA) at a concentration of 100 nM) using Lipofectamine 2000 (Invitrogen via ThermoFisher Scientific, Waltham, MA, USA) according to the manufacturer’s instructions. A random sequence was used as negative control, (miRNA MimicNegative Control #1, ThermoFisher Scientific, Waltham, MA, USA). After 4 h of transfection, media was replaced with podocyte culture medium, as above, and cells were further incubated for 48 h.

Human glomerular endothelial cells were seeded 2 days prior to transfection. The same transfection procedure as described above was also used for human glomerular endothelial cells. Cell viability and morphology were determined on an optical microscope before and after every treatment.

hsa-mir-200c-3p sequence: UAAUACUGCCGGGUAAUGAUGGA.

#### 4.1.2. RNA Isolation, cDNA Transcription, and qPCR

RNA from whole cell lysates was isolated using the ReliaPrepTM RNA Miniprep System (Promega, Madison, WI, USA), according to the manufacturers’ protocol. For mRNA reverse transcription, 1 μg of RNA was transcribed using the following reagents: 5× RT buffer and M-MLV RT 50.000 U (Promega, Madison, WI, USA), dNTP Mix (Promega, Madison, WI, USA), random hexamer primer (ThermoFisher Scientific, Waltham, MA, USA), and RiboLock (Thermo Fisher Scientific, Waltham, MA, USA). Reverse transcription was carried out at 25 °C for 5 min, 40 °C for 60 min and 70 °C for 10 min. Sybr green-based real-time PCR (Maxima SYBR Green/ROX qPCR Master Mix, Thermo Fisher Scientific, Waltham, MA) was performed with the following protocol: 10 min at 95 °C followed by 40 cycles of 15 s at 95 °C and 1 min at 60 °C, followed by 15 s at 95 °C, 1 min at 60 °C and 15 s at 95 °C. Individual samples were run in triplicates. The following primers were used

hGAPDH fw: 5′-GTCAGCCGCATCTTCTTTTG-3′hGAPDH rev: 5′-GCGCCCAATACGACCAAATC-3′hHPRT fw: 5′-GACCAGTCAACAGGGGACAT-3′hHPRT rev: 5′-AACACTTCGTGGGGTCCTTTTC-3′hVEGF-A fw: 5′-CAACAAATGTGAATGCAGACCAAA-3′hVEGF-A rev: 5′-CCCTTTCCCTTTCCTCGAACT-3′hZEB1 fw: 5′-CCAGCCAAATGGAAATCAGGATG-3′hZEB1 rev: 5′-TGGGCGGTGTAGAATCAGAG-3′hZEB2 fw: 5′-CCTCTGTAGATGGTCCAGAAGA-3′hZEB2 rev: 5′-AATTGCGGTCTGGATCGTGG-3′zvegf-aa fw: 5′-CAAGTGTGAATGCAGGCCAAA-3′zvegf-aa rev: 5′-AGCATTTACAGGTGAGGGGG-3′zvegf-ab fw: 5′- GAATGCCAGTGTCGGATGAAA-3′zvegf-ab rev: 5′-TAGCTGAAGTTAGGCAGGATGG-3′zhprt fw: 5′-ACCAAAACACTATGCGGCTG-3′zhprt re: 5′-GTGTCCACCCATGTCCTTCA-3′

Data were analyzed by using the ΔΔCt method.

#### 4.1.3. MiR Isolation, cDNA Transcription and qPCR

The isolation of miRs was performed by using the miRNeasy Mini Kit (Qiagen, Hilden, Germany) and cDNA transcription. qPCR was performed using TaqMan probes for microRNA: hsa-miR-200c-3p and U6 control miR (Thermo Fisher Scientific, Waltham, MA, USA) according to the manufacturer’s protocol. Data were analyzed by using the ΔΔCt method.

#### 4.1.4. Immunofluorescence Microscopy

Glomerular endothelial cells or podocytes were grown on uncoated glass cover slips (Menzel–Glaser, via VWR International GmbH, Darmstadt, Germany) and were transfected as above. After 48 h of transfection, cells were fixed with 4% paraformaldehyde for 10 min and permeabilized with 0.5% Triton. Cells were blocked with 10% sheep serum (abcam, Cambridge, UK), 1% BSA and 0.5% Triton for 1 h at room temperature. Afterwards, cells were incubated with the primary antibodies diluted in 3% sheep serum/1% BSA/Triton 0.5% at 4 °C for overnight. 

The dilutions used for the primary antibodies are as follows: for anti-ZEB1 (NBP1-05987, Novus Biologicals, via biotechne, Minneapolis, MN, USA) 1:100, for VEGF-A (Sc-507, Santa Cruz Biotechnology Inc., Heidelberg, Germany) 1:200. The cells were then washed with PBS and further incubated with secondary antibodies for 1 h at room temperature in the dark using donkey anti-rabbit Alexa Flour 488 (Invitrogen, via ThermoFisher Scientific, Madison, WI, USA) secondary antibody for ZEB, donkey anti-mouse Alexa 488 green (Invitrogen, via ThermoFisher Scientific, Madison, WI, USA) for VEGF-A. Actin filaments were stained with phalloidin (Alexa Flour 555, Invitrogen, via ThermoFisher Scientific, Madison, WI, USA) at a dilution of 1:400 for 20 min at room temperature in the dark. Cell nuclei were stained with Hoechst (Sigma-Aldrich Chemie GmbH, Taufkirchen, Germany). Cover slides were mounted with fluorescence mounting medium (ThermoFisher Scientific, Madison, WI, USA) and imaged using fluorescence microscopy (Leica DMI 4000, Leica Microsystems GmbH, Wetzlar, Germany).

### 4.2. ELISA

Following miR-200c overexpression in podocytes, cell culture supernatants were measured for VEGF protein using VEGF Human ELISA Kit (Invitrogen, via ThermoFisher Scientific, Madison, WI, USA) according to the manufacturer’s protocol.

### 4.3. Zebrafish Experiments

Zebrafish strains: wild-type AB (ZIRC, Eugene, OR, USA), transgenic Tg(l-fabp:VDBP-EGFP) and transgenic wt1b:EGFP. All strains were grown and mated at 28.5 °C, and embryos were kept and handled in standard E3 solution. mirVana^®^ negative control and mirVana^®^ miRNA mimic, hsa-miR-200c-3p (ThermoFisher Scientific, Madison, WI, USA), were injected in one- to four-cell stage of the zebrafish larvae using a Nanoject II injection device (Drummond Scientific, Broomall, PA, USA) in injection buffer (100 mM KCl, 0.1% phenol red). At 48 hpf, remaining chorions were manually removed. The MDI Biological Laboratory IACUC (#11-02) approved the animal protocol.

#### Zebrafish Assay for Proteinuria

Tg(l-fabp:VDBP:EGFP) zebrafish, which express a green fluorescent vitamin D-binding protein (VDBP-EGFP), under the control of the liver-type fatty acid-binding protein (l-fabp) promoter were used to screen for proteinuria in zebrafish larvae. The green fluorescence of the fish can easily be detected in the retinal plexus of the fish with fluorescence microscopy. The VDBP-EGFP fusion protein has a molecular weight of approximately 78 kDa. If the GFB is impaired, the fish loses plasma proteins, and the eye fluorescence decreases. Injected miRNA was titrated in dose response experiments. Between 20 and 40 fish were alive in each group 120 hpf. Results shown depict a representative experiment out of three independent injection experiments performed.

### 4.4. Transmission Electron Microscopy

Followed by miR-200c injection, zebrafish larvae were fixed in solution D overnight, washed 3x in 0.1 M cacodylate buffer and postfixed in 1% OsO_4_ for 1 h. Tissues were once again washed, dehydrated and embedded in EPON (recipe/protocol from EMS, Hatfield, PA, USA). Semi-thin (300 nm) and ultra-thin (90 nm) sections were cut with a Leica UC-6 Microtome (Leica Microsystems GmbH, Wetzlar, Germany). Sections obtained were mounted onto formvar-coated Ni slot grids (EMS, Electron Microscopy Sciences, Hatfield, PA, USA). Grids were stained for 30 min in 5% uranyl acetate (Science Services, München, Germany), followed by 0.1% lead citrate (Science Services, München, Germany) for 15 min.

### 4.5. Statistics

All data are shown as mean ± SEM and were compared by ANOVA or T-test to analyze for statistical significance. Experiments were performed at least three times.

## Figures and Tables

**Figure 1 ijms-23-15070-f001:**
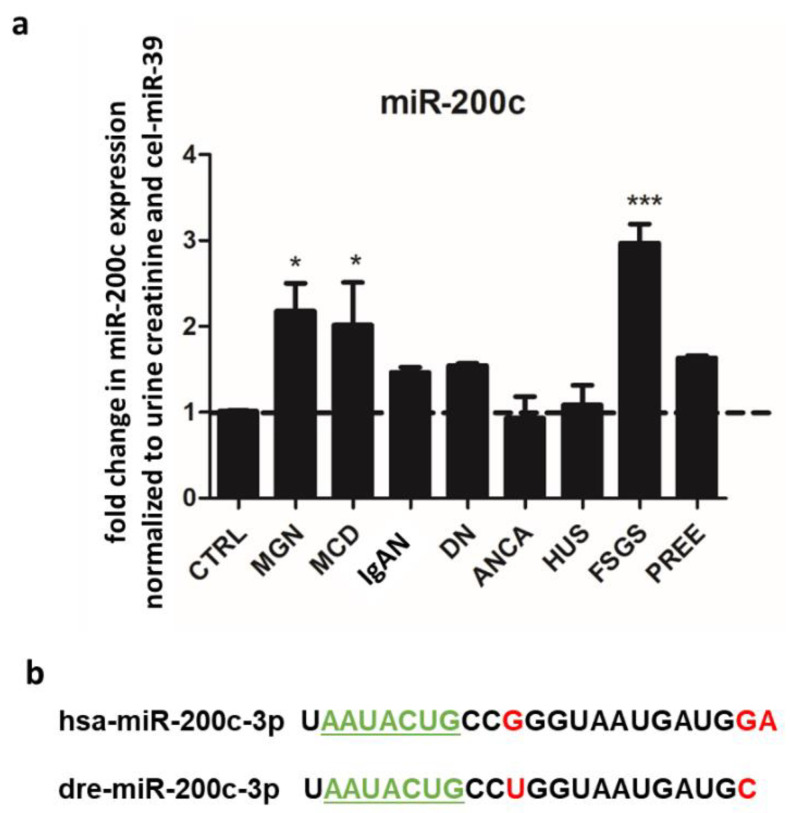
miR-200c is upregulated in urine from patients with MGN, MCD and FSGS. (**a**) Fold change in urinary miR-200c expression normalized to urine creatinine and cel-39 in different renal diseases. N = 27 different patients. * *p* < 0.05, *** *p* < 0.001 compared to control. CTRL: control, MGN: membranous glomerulonephritis, MCD: minimal change disease, IgAN: IgA nephropathy, DN: diabetic nephropathy, ANCA: antineutrophil cytoplasmic autoantibody vasculitis, HUS: hemolytic uremic syndrome, FSGS: focal segmental glomerulosclerosis, PREE: preeclampsia. (**b**) Sequence of human (hsa) and zebrafish (dre) miR-200c. Differences are highlighted in red. The seed region of the miRs are underlined and labeled in green.

**Figure 2 ijms-23-15070-f002:**
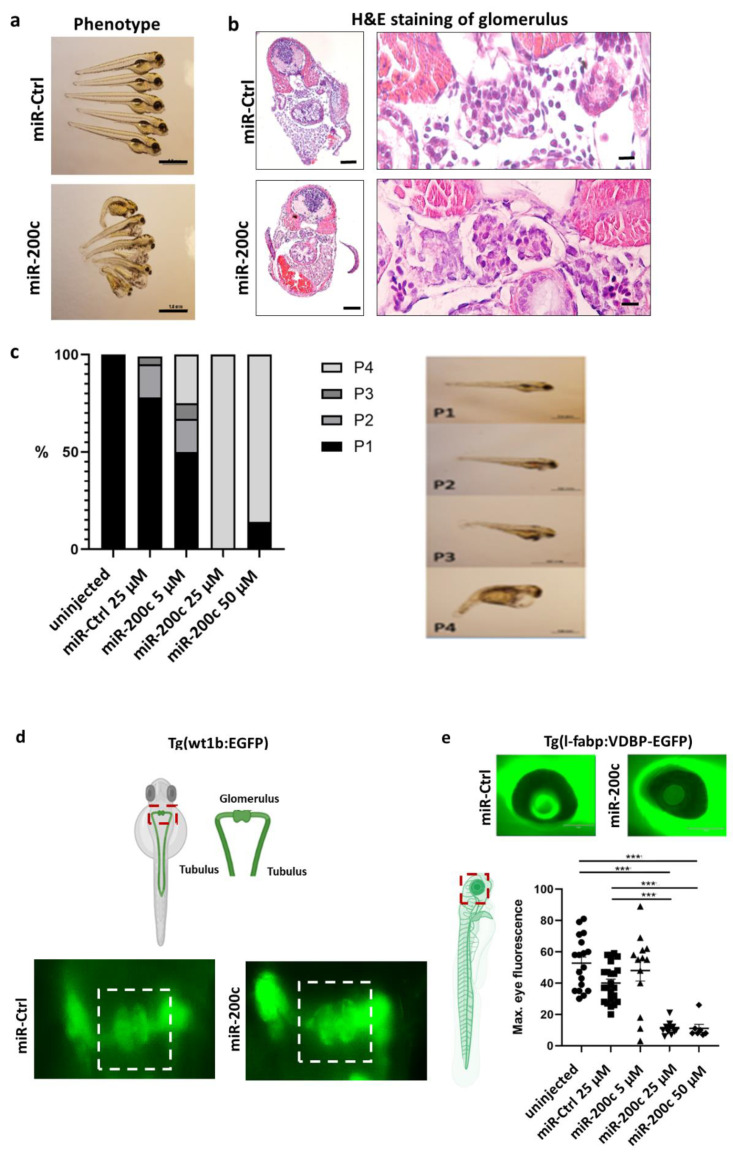
miR-200c mimic induces edema, proteinuria, podocyte effacement and loss of endothelial fenestrations. Zebrafish were injected with an miR-Ctrl mimic and miR-200c-3p mimic at the one-to-four cell stage. (**a**) Phenotype pictures of zebrafish larvae at 96 h postfertilization (hpf). Scale bar = 500 µm. (**b**) Staining the zebrafish pronephroi with hematoxylin–eosin (H&E) at 96 hpf after miR-Ctrl and miR-200c mimic injection in egg stage. Scale bar = 100 µm and 10 µm in the zoom in (**c**) Quantification of phenotype by severity of pericardial and yolk sac edema in percentage. Examples of phenotypic changes, grading the normal zebrafish phenotype with no edema P1 and the most “edematous” phenotype with P4 at 96 hpf. (**d**) Timely fusion of the glomeruli at 48 hpf after miR-200c overexpression in Tg (wt1b:EGFP) zebrafish, which express eGFP under the control of the wt1b promotor. Scale bar = 200 µm. (**e**) Representative images of an eye of Tg(I-fabp:VDBP:EGFP) fish, which were injected with an miR-Ctrl or miR-200c mimic. Scale bar = 150 µm. Quantification of loss of circulating high-molecular-weight proteins by measuring max. eye fluorescence in the retinal vessel plexus of Tg(l-fabp:VDBP:EGFP) zebrafish larvae at 96 hpf. *** *p* < 0.001. (**f**) Top: TEM pictures of the zebrafish GFB at 96 hpf after miR-200c mimic and miR-Ctrl injection in egg stage. White arrows show fenestration in the control zebrafish, which are lost in miR-200c-injected zebrafish. Black arrows show podocyte effacement. Scale bar = 500 nm. Bottom: quantification of loss of endothelial fenestrae and podocyte effacement in percentage of representative fish in each group. * *p* < 0.05; *** *p* < 0.001.

**Figure 3 ijms-23-15070-f003:**
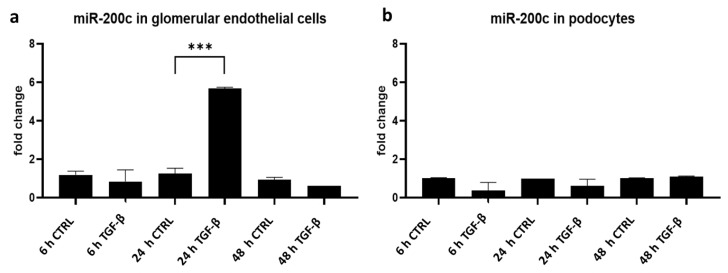
TGF-β stimulation induces miR-200c expression in cultured glomerular endothelial cells. (**a**,**b**) qPCR for miR-200c detection in cultured human glomerular endothelial cells (**a**) and human podocytes (**b**) normalized to U6 RNA. Data are given as fold change compared to unstimulated cells (CTRL) at the respective time point. Cells were stimulated with TGF-β for 6 h, 24 h and 48 h, *** *p* < 0.001. Reproducible data from three independent experiments.

**Figure 4 ijms-23-15070-f004:**
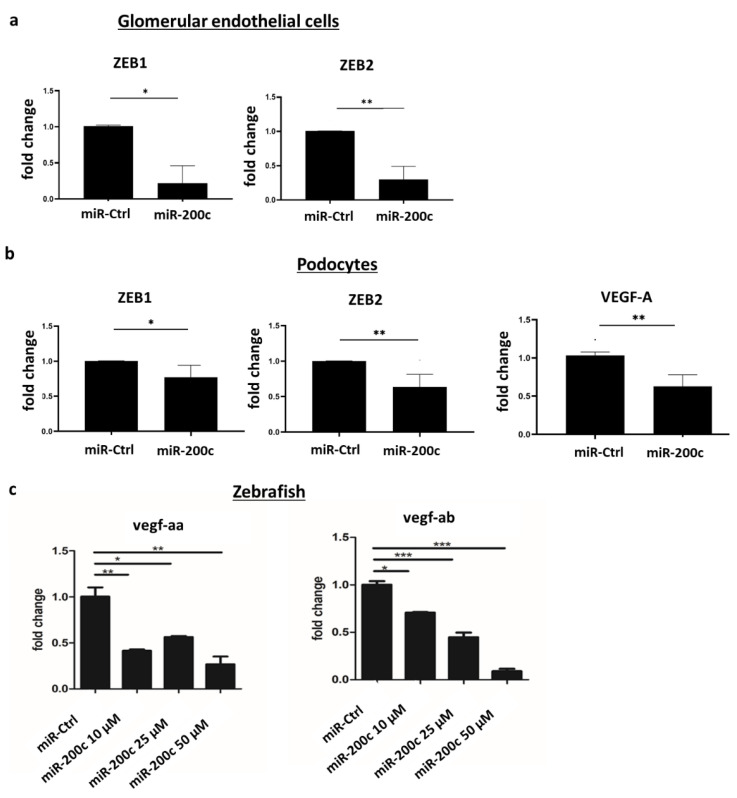
miR-200c overexpression alters the expression of ZEB1/2 and VEGF-A. (**a**,**b**) Cultured human glomerular endothelial cells and podocytes were transfected with an miR-control (miR-Ctrl) or miR-200c mimic (miR-200c), and qPCR was performed. (**a**) miR-200c-induced downregulation of ZEB1/2 in glomerular endothelial cells. (**b**) miR-200c-induced downregulation of ZEB1/2 and VEGF-A in podocytes. * *p* < 0.05, * *p* < 0.01, n = 4 independent experiments. (**c**) qPCR showing vegf-aa and vegf-ab mRNA expression after injection of miR-Ctrl or different concentrations of miR-200c mimic in zebrafish. Injections were performed at one two four cell stage. qPCR was performed 120 hpf. * *p* < 0.05, ** *p* < 0.01, *** *p* < 0.001. A pool of 20 zebrafish was used for RNA isolation in each group.

**Figure 5 ijms-23-15070-f005:**
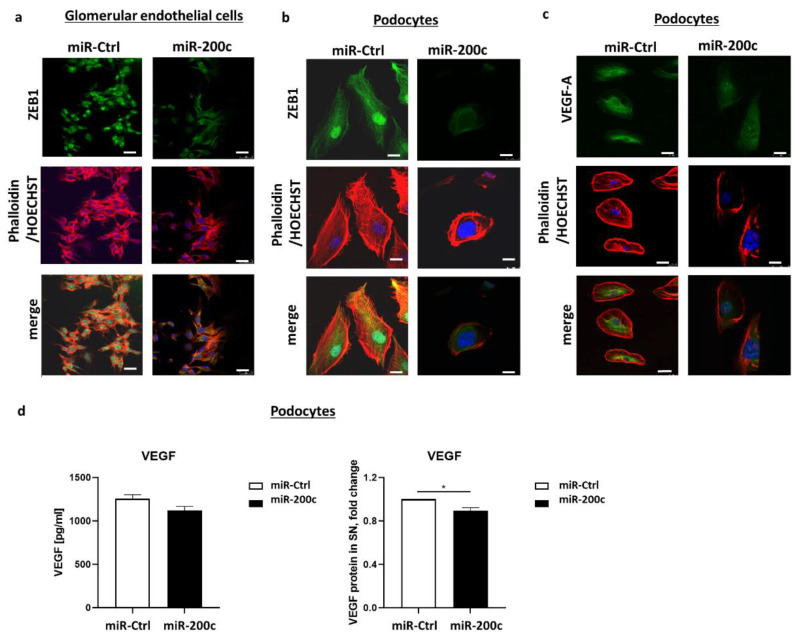
miR-200c overexpression decreases VEGF-A and ZEB1 on protein level. Images show immunofluorescent staining with phalloidin (red), anti-ZEB1 (green in (**a**,**b**)) and anti-VEGF-A (green in (**c**)). Nuclear staining is shown in blue with Hoechst. Scale bar = 25 µm. (**d**) VEGF protein in supernatant of cultured human podocytes (left panel shows total concentration, right panel shows fold change) after transfection with miR-200c mimic or miR-Ctrl was measured with ELISA. * *p* < 0.05.

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
