# Peer review of "Glomerular Endothelial Cell-Derived miR-200c Impairs Glomerular Homeostasis by Targeting Podocyte VEGF-A"

_ijms, 2022, doi:10.3390/ijms232315070_

Round 1
Reviewer 1 Report
This study by Ursu et al examined the role of miR-200c in the cross talk between glomerular endothelial cells and podocytes and the mediating effect of VEGF-A by analyzing human urine samples, zebrafish and cultured human podocytes.
Overall, this is a carefully designed and well conducted study involving both in vitro and in vivo model systems of podocyte pathobiology.
The manuscript could be improved by addressing the following issues.
1. In Figure 1, was the urinary levels of miR-200c corrected by urine creatinine levels? What the sample size (n=?)? Why there is no SD? It seems that miR-200c is upregulated in urines from patients with most types of podocytopathies. However, why its level is decrease in patients with FSGS? This should be properly explained in order to justify the central model proposed by the authors.
2. Is the sequence of miR-200c highly conserved in zebrafish? Please examine the expression of VEGF-A in the zebrafish experiment.
Author Response
Dear reviewer,
you can find our comments in the attached word file.
Thank you very much.
Janina Müller-Deile

Reviewer 2 Report
Ursu et al show that Glomerular endothelial cell-derived miR-200c impairs glomerular homeostasis by targeting podocyte VEGF-A. They show increased miR-200c in urine of patients with MCD, MGN and preeclampsie. they also show miR-200c overexpression causing proteinuria, edema, podocyte effacement and glomerular endotheliosis in zebrafish, moreover the authors show in cultured cells that 24h of TGF-beta treatment increases miR-200c expression in glomerular endothelial cells and not in podocytes. Overall the paper is well written and the data support the authors idea.
Author Response
Dear reviewer,
thank you very much for your feedback on our article " Glomerular endothelial cell-derived miR-200c impairs glomerular homeostasis by targeting podocyte VEGF-A" which helped us to improve our manuscript.
We appreciate that you liked our article.
Best
Janina Müller-Deile
Round 2
Reviewer 1 Report
The authors have adequately addressed my comments.